# Different depths of sedation versus risk of delirium in adult mechanically ventilated patients: A systematic review and meta-analysis

**Ling Long[1], Shan Ren[1], Yichun Gong[2], Haotian Zhao[3], Cong He[1], Limin Shen[1], Heling Zhao[1]☯, Penglin Ma[iD][1,4]☯ ***

**1** Department of Intensive Care Unit, Hebei General Hospital, Shijiazhuang City, Hebei Province, China, **2** SICU, the 8th Medical Center, General Hospital of PLA, Beijing City, China, **3** Ultrasound Department, Hebei General Hospital, Shijiazhuang City, Hebei Province, China, **4** Critical Care Medicine Department, Peking University Third Hospitial, Beijing City, China

☯ These authors contributed equally to this work.

* mapenglin1@163.com

**Funding:** There are no funding.

## Abstract

### Background

Delirium is multifactorial. This study aimed at determining the association between different depths of sedation and the risk of delirium in adult mechanically ventilated patients.

### Methods

A systematic literature retrieval was conducted in databases including Cochrane Central Register of Controlled Trials, PubMed, Embase, Cumulative Index to Nursing and Allied Health Literature for publications available till December 2019 without limitation in study type, and followed by a secondary retrieval for related literature. STATA15.1 and WinBugs 14.3 were used in statistical analyses for different sedation depths as the intervention. The main endpoint was delirium occurrence. Secondary endpoints were agitation-related adverse events and mortality.

### Results

We included 18 studies comprising 8001 mechanically ventilated patients. Different sedation depths were not associated with the occurrence of delirium (OR = 1.00, 95%CI: 0.64–1.58, $P = 0.993$). Among the 18 enrolled studies, this finding was not confounded by the dosage of benzodiazepines (OR = 0.96, 95%CI: 0.79–1.17, $P = 0.717$) in eight randomized controlled trials(RCTs) or the patients' disease severity(OR 0.95, 95%CI: 0.79–1.13, $P = 0.548$) in 10 RCTs. However, contrasting results were found in non-RCTs. The deeper sedation group had a significantly increased risk for death(OR = 1.82, 95% CI: 1.23–2.69, $P = 0.003$), whereas lighter sedation seemed a potential risk for agitation-related adverse events (OR = 0.61, 95%CI: 0.45–0.84, $P = 0.002$).

**Competing interests:** The authors have declared that no competing interests exist.

## Conclusions

It is inconclusive whether significantly different sedation depths would change the risk of delirium in adult mechanically ventilated patients.

## Trial registration number

The study was registered in PROSPERO(http://www.crd.york.ac.uk/PROSPERO/) under registration number CRD42019145276.

## Introduction

Delirium is an acute brain dysfunction [1]. Previous studies demonstrated that the incidence of delirium was approximately 10%-30% in hospitalized patients and 80% in critically ill patients under mechanical ventilation (MV) [2,3]. Notably, increasing evidence suggests that delirium was associated with poor outcomes of ICU patients such as ICU readmission and prolonged hospitalization, long-term cognitive impairment and high mortality [2,4].

The mechanism of delirium remains unclear [5]. Risk factors for delirium include illness-related acute pathophysiological abnormalities(e.g., hypotension, acidosis, hypoxia and sepsis) [6], environmental factors (e.g., lighting, alarm sounds, and noise); and iatrogenic harms (e.g., frequent suctions, puncture, immobilization and even use of analgesic and sedative drugs against these stimuli) [7–9]. Of those, potential and modifiable risk factors were highly interested, for example minimizing sedation and against use of benzodiazepines [10]. Significantly, numerous studies reported that patients receiving deep sedation were more susceptible to post-traumatic stress disorder syndrome, ICU memory disorder and delirium [11,12]. A recent published meta-analysis revealed that delirium frequency was 28.7% in the light sedation group in comparison with 48.5% in the deep sedation group of patients with MV, but odds ratio, 0.50 (0.22–1.16) [13]. It was presumed that limited data from randomized controlled trials (RCT) was inadequately powered to show a significant association while high heterogeneity existed in the enrolled studies. In fact, there were a few published RCTs primarily regarding the effect of sedation depth on delirium occurrence in critically ill patients with MV yet [14,15]. Meanwhile, it was noticed that a significant difference of sedation depths (e.g., lighter vs. deeper) as a component of intervention was involved in the design of some RCTs [16–19]. Interestingly, these studies compared clinical outcomes such as the occurrence of delirium, agitation-related adverse events, and mortality. Therefore, we hypothesized that adding additional high quality data extracted from these studies, an updated meta-analysis might provide a more convincible conclusion on the relationship between sedation depth and the occurrence of delirium in critically ill patients under MV.

## Methods

A systematic review protocol was prepared in accordance with the Preferred Reporting Items for Systematic Reviews and Meta-Analysis Protocol (PRISMA-P) statement. This systematic review was registered in the PROSPERO international prospective register of systematic reviews (CRD42019145276). Ethical approval was vaived for this study.

## Study identification

The literature retrieval strategy followed the Cochrane Interventional Systems Review Interventions [20], by which we searched databases, including Cochrane Central Register of Controlled Trials (2019, Issue12), Pubmed (from 1946 to December 31, 2019), Embase (from 1974 to December 31, 2019), and Cumulatve Index to Nursing and Allied Health Literature (from 1937 to December 31, 2019). We also conducted a secondary retrieval for selected literature (S1 File describes the retrieval strategy details). Our search strategy was prepared with the assistance of a medical librarian.

## Inclusion criteria

**Patients.** Adult patients (age >18 years) under MV were eligible. Patients with alcohol withdrawal syndrome, brain injury or speech disorders were excluded because these factors may confound the evaluation of delirium.

**Intervention.** Depth of sedation. The sedation depth in these studies was evaluated primarily by using sedation score scales [e.g., Richmond Agitation-Sedation Scale (RASS) [21], Riker Sedation-Agitation Scale (SAS) [22], Ramsay Sedation scale [23], and Motor Activity Assessment Scale (MAAS) [24]]. There were no strict limitations on category, dosage, or administration method or rate of the sedative infusion. Studies involving different sedation depths between two groups were eligible, including studies with the sedation depth clearly defined in their methodology, based on which patients were grouped (e.g., deep sedation vs. light sedation); studies without such a definition but that implied statistical difference in sedation depth (P<0.05); and studies using sedation as a component of intervention and showing a difference in sedation depth (P<0.05). In this meta-analysis, the terms of lighter sedation and deeper sedation were predefined as the intervention for classification of two groups in the enrolled studies.

**Comparisons.** The groups compared were deep sedation versus light sedation, deeper sedation versus lighter sedation, sedation versus no sedation, and routine sedation versus prevent deep sedation.

**Outcomes.** The main outcome was the occurrence of delirium. The diagnosis of delirium was based primarily on the Confusion Assessment Method for the ICU (CAM-ICU) scale [25] and the Diagnostic and Statistical Manual of Mental Disorders, 4th Edition (DSM-IV) [26]. Secondary outcomes included the incidence of agitation-related adverse events (self-extubation), and mortality.

**Document types.** Randomized controlled studies, before-after controlled studies, and prospective/retrospective cohort studies were included. Animal experiments, reviews, case reports and studies with incomplete data were excluded.

## Study selection and data abstraction

Based on the objective of the study, the required data and information were listed. Two researchers independently reviewed all literature. All documents were initially de-duplicated, and then screened based on the title and content of the abstract. If the eligibility of a literature report was difficult to determine, the full text was retrieved for further screening with the reasons for exclusion, and it was documented in the inclusion/exclusion flow chart. A crosscheck was then conducted. Any discrepancy was settled by discussion or consultation with a third evaluator.

An electronic data sheet was established, based on the requirements of the *Cochrane Handbook for Systematic Reviews of Interventions*, which includes the general data of the included literature (e.g., author, year, and experimental design method), the enrolled patients' profiles

(e.g., sample size, and severity of disease), intervention characteristics (e.g., depth of sedation), and indicators of outcome (e.g., delirium occurrence, incidence of agitation-related adverse events, and mortality). In instances of incomplete data from the included literature or any doubt existed regarding the data, the study's author was consulted. Two authors independently extracted the data included in the literature. Differences were resolved by consensus or by consulting a third author.

## Study quality assessment

The two authors used the biased risk assessment tool described in the Cochrane Manual to evaluate the risk of the included randomized controlled studies [20]. They used the Newcastle-Ottawa Scale to evaluate the risk of bias assessment for nonrandomized controlled studies [27]. Final decisions regarding differences were determined by discussion or consultation with a third author. In instances of missing data, the original author was contacted to retrieve the data. If the author was out of contact, the research team explored the reasons for the missing data and its possible impact on the results. If a sufficient number of studies (e.g., more than 10 studies) were eligible for inclusion, a funnel chart was used to assess publication bias. If the funnel charts seemed asymmetric, the team discussed possible causes, which was then confirmed by statistical tests.

## Data analysis

Statistical softwares STATA15.1 (StataCorp., College Station, TX, USA) and WinBugs14.3 (Imperial College and Medical Research Council, London, UK) were applied to evaluate the efficacy of intervention. For bicategorized data (e.g., delirium occurrence, incidence of agitation-related adverse events, and mortality), we used the odds ratio (OR) and 95% confidence interval (95%CI). We also explored the clinical heterogeneity, based on differences in research design, population profiles, and intervention measures.

We also used STATA15.1 software to analyze clinical heterogeneity and methodological heterogeneity, and to evaluate statistical heterogeneity by calculating $Chi^2$ or $I^2$ statistics. First, heterogeneity among the studies was determined using $Chi^2$, the significance level was $P = 0.10$. To quantitatively analyze heterogeneity, $I^2$ was used; the significance level was 50%. For $P > 0.1$ and $I^2 < 50\%$, homogeneity could be considered in multiple similar studies, and a fixed-effect model was applied to their meta-analysis. For $P < 0.1$ and $I^2 \geq 50\%$ but with the need for combination, based on the intergroup consistency (determined clinically), random effect models were selected for the meta-analysis [20]. When high heterogeneity was detected among the included studies, a subgroup analysis or Bayesian method exploration was conducted to minimize the impact of the heterogeneity on the results.

## Results

Based on the retrieval strategy, 408 literature reports were identified; of these, 130 reports were excluded because of duplication, 158 reports were excluded after reviewing their titles and abstracts, and 105 reports were excluded after reviewing their full text. A second retrieval for the selected literature reports was conducted by reviewing their references and other related literature. We ultimately enrolled 18 studies reporting a correlation between different sedation depths and risk of delirium [14–19, 28–39], that included 10 RCTs, four before-after controlled trials, two prospective cohort studies, and two retrospective studies (Fig 1). Table 1 presents the general profiles of the studies enrolled. Based on the analysis of potential confounding factors influencing the occurrence of delirium, 13 of 18 studies indicated a significant difference in the dosage of benzodiazepines between deeper and lighter sedation. Moreover, 13 studies

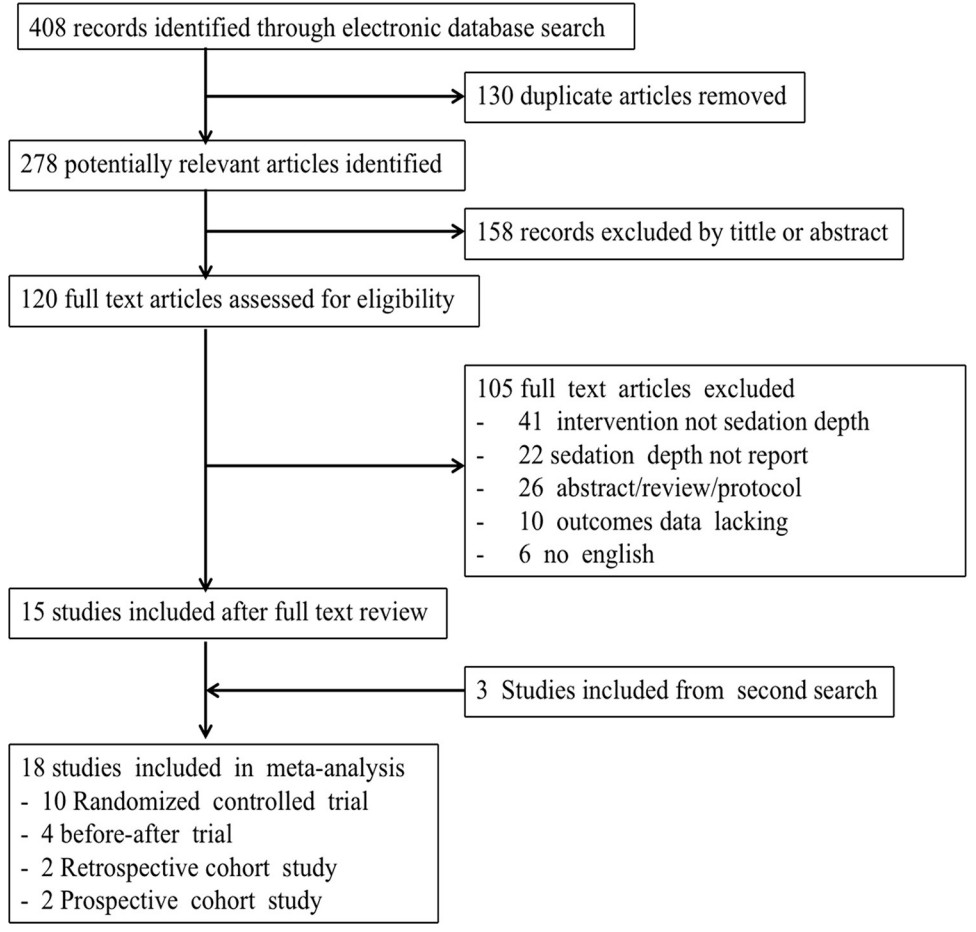

**Fig 1. Flowchart of the recruited studies.**

similarly reported no statistically significant difference in disease severity of patients between two sedation groups (S1 Table).

Using Cochrane Risk of Bias Tool for randomized controlled studies and the Newcastle-Ottawa Scale for nonrandomized controlled studies indicated a low bias risk in all 18 studies (S2 Table, S1 Fig and Fig 2). A funnel chart for the occurrence of delirium showed a symmetrical distribution of the relative risk ratio of the occurrence of delirium ($P = 0.711$), which indicated a lower risk of publication bias. No study with a high risk of bias was discovered in the sensitivity analysis (Fig 3 and S2 Fig).

## Correlation between sedation depth and delirium occurrence

In the meta-analysis, we included 8001 patients under MV from the 18 studies. The delirium occurrence was not significantly increased in patients under ventilation with deeper sedation, compared with patients with lighter sedation (OR = 1.00, 95%CI:0.64–1.58, $P = 0.993$; Fig 4). A high interstudy heterogeneity existed ($I^2 = 93.2\%$, $P<0.001$). Non-RCTs was the main reason for the high heterogeneity, based on subgroup analysis ($I^2 = 96.8\%$, $P<0.001$). Cohort studies by Shehabi, van den Boogaard and Balzer showed that deeper sedation increased the delirium risk [28,29,34]. However, three studies [30,33,35] reported an even higher delirium occurrence with lighter sedation. Despite low heterogeneity ($I^2 = 0.0\%$, $P = 0.771$) and high mergeability,

**Table 1. Characteristics of the included studies.**

| Author/Year | Study design | $N_D/N_L$ | Age | Different BDZ use | Different SI | Sedation depth | Delirium assessment | Outcomes | Endpoints (primary/secondary) |
|---|---|---|---|---|---|---|---|---|---|
| Pandharipande et al. [17] (2007) | RCT | 51/52 | 59/60 | Yes | No | RASS≤-3 | CAM-ICU | ①②③ | 1. Duration of brain organ dysfunction (delirium or coma) |
| | | | | | | | | | 2. Prevalence of brain organ dysfunction (delirium or coma) |
| | | | | | | RASS≥-2 | | | 3. Mechanical ventilator-free day |
| | | | | | | | | | 4. Intensive care unit length of stay |
| | | | | | | | | | 5. Mortality(28-day) |
| | | | | | | | | | 6. Self-extubation |
| | | | | | | | | | 7. Cost of care |
| Samuelson et al. [14]. (2008) | RCT | 18/18 | 66/66 | No | No | MAAS 1 to 2 | CAM-ICU | ① | 1. ICU length of stay |
| | | | | | | | | | 2. Post-operative complications |
| | | | | | | MAAS 3 to 4 | | | 3. Re-intubated |
| | | | | | | | | | 4. Duration of mechanical ventilation |
| | | | | | | | | | 5. Delirium |
| Girard et al. [16] (2008) | RCT | 168/167 | 64/60 | No | No | RASS -2.5 | CAM-ICU | ①②③ | 1. Time breathing without assistance |
| | | | | | | | | | 2. Length of stay in ICU and hospital |
| | | | | | | RASS -1.0 | | | 3. Delirium |
| | | | | | | | | | 4. Complication (self-extubation) |
| | | | | | | | | | 5. Mortality (28-day and 1-year) |
| Treggiari et al. [15] (2009) | RCT | 50/52 | 60/63 | Yes | No | Ramsay 3 to 4 | DSM-IV | ①②③ | 1. PTSD-related symptoms |
| | | | | | | | | | 2. Mortality (ICU and hospital) |
| | | | | | | | | | 3. Duration of mechanical ventilation |
| | | | | | | Ramsay 1 to 2 | | | 4. Length of stay in ICU and hospital |
| | | | | | | | | | 5. Incidence of organ dysfunction |
| | | | | | | | | | 6. Complication (self-extubation) |
| Strøm et al. [18] (2010) | RCT | 58/55 | 65/67 | Yes | No | Ramsay 3 to 4 | DSM-IV | ①②③ | 1. Mechanical ventilator-free days |
| | | | | | | | | | 2. Length of stay in ICU and hospital |
| | | | | | | No sedation | | | 3. Mortality (ICU and hospital) |
| | | | | | | | | | 4. Delirium |
| | | | | | | | | | 5. Complication (self-extubation) |
| Shehabi et al. [28] (2012) | Prospective cohort study | 171/80 | Unclear | No | Unclear | RASS -5 to -3 | CAM-ICU | ①③ | 1. Time to extubation |
| | | | | | | | | | 2. Time to delirium |
| | | | | | | RASS -2 to 1 | | | 3. Time to hospital mortality |
| | | | | | | | | | 4. Mortality(180-day) |
| van den Boogaard et al. [29] (2012) | Prospective cohort study | 249/1017 | Unclear | No | Unclear | RASS-4 to -3 | CAM-ICU | ①③ | 1. Duration of mechanical ventilation |
| | | | | | | RASS≥-2 | | | 2. Reintubation |
| | | | | | | | | | 3. Unplanned tube or catheter removal |
| | | | | | | | | | 4. Length of stay in ICU |
| | | | | | | | | | 5. Mortality(hospital) |

*(Continued)*

**Table 1.** (Continued)

| Author/Year | Study design | $N_D/N_L$ | Age | Different BDZ use | Different SI | Sedation depth | Delirium assessment | Outcomes | Endpoints (primary/secondary) |
|---|---|---|---|---|---|---|---|---|---|
| Shehabi et al. [19] (2013) | RCT | 16/21 | 61/65 | Yes | No | RASS-5 to -3 | CAM-ICU | ①②③ | 1. Number of RASS measurements between –2 and –3 in 48 hour |
| | | | | | | RASS-2 to 1 | | | 2. Time to randomization |
| | | | | | | | | | 3. Dose and duration of rescue sedatives and opioids |
| Hager et al. [30] (2013) | Before-after trial | 120/82 | 48/52 | Yes | No | RASS-5 to -2 | CAM-ICU | ① | 1. Dose and duration of sedatives |
| | | | | | | RASS-3 to 0 | | | 2.Delirium |
| Shehabi et al. [31] (2013) | RCT | 30/30 | 48/52 | Yes | No | RASS<-2 | CAM-ICU | ①③ | 1. Number of RASS measurements between –2 and –3 in 48 hour |
| | | | | | | RASS-2 to 1 | | | 2. Time to randomization |
| | | | | | | | | | 3. Dose and duration of rescue sedatives and opioids |
| Nassar et al. [32] (2014) | RCT | 30/30 | 51/47 | Yes | No | SAS 2.6 to 3.7 | CAM-ICU | ①②③ | 1. Mechanical ventilator-free days |
| | | | | | | | | | 2. Mortality (ICU and hospital) |
| | | | | | | SAS 3.4 to 4.0 | | | 3. Delirium |
| | | | | | | | | | 4. Nurse workload |
| | | | | | | | | | 5. Self-extubation and psychological distress |
| Dale et al. [33] (2014) | Before-after trial | 703/780 | 48/49 | Yes | No | RASS-1.30 | CAM-ICU | ①③ | 1. duration of mechanical ventilation. |
| | | | | | | | | | 2. delirium |
| | | | | | | RASS-0.99 | | | 3. Frequency of assessment with the RASS and CAM-ICU instrument |
| | | | | | | | | | 4. Benzodiazepine dosing |
| | | | | | | | | | 5. Length of stay in ICU and hospital |
| | | | | | | | | | 6. Mortality(hospital) |
| Balzer et al. [34] (2015) | Retrospective cohort study | 513/1371 | 64/68 | Yes | Yes | RASS -5 to -3 | CAM-ICU | ①③ | 1. Mortality (ICU and hospital) |
| | | | | | | | | | 2. Length of stay in ICU and hospital |
| | | | | | | RASS -2 to 0 | | | 3. Time to extubation |
| | | | | | | | | | 4. Delirium |
| | | | | | | | | | 5. Hemodialysis in first 48 hour |
| Skrupky et al. [35] (2015) | Before-after trial | 97/102 | 58/56 | Yes | No | RASS-5 to -3 | CAM-ICU | ①②③ | 1. Level of sedation |
| | | | | | | | | | 2. Delirium |
| | | | | | | | | | 3. Duration of mechanical ventilation |
| | | | | | | | | | 4. Length of stay in ICU and hospital |
| | | | | | | RASS-2 to 0 | | | 5. Mortality |
| | | | | | | | | | 6. Complication (self-extubation) |

(*Continued*)

**Table 1.** (Continued)

| Author/Year | Study design | $N_D/N_L$ | Age | Different BDZ use | Different SI | Sedation depth | Delirium assessment | Outcomes | Endpoints (primary/secondary) |
|---|---|---|---|---|---|---|---|---|---|
| Stephens et al. [36] (2017) | Retrospective cohort study | 231/ 132 | Unclear | No | Unclear | RASS-5 to -3 | CAM-ICU | ①③ | 1. Mortality(hospital) |
| | | | | | | | | | 2. Duration of mechanical ventilation |
| | | | | | | RASS>-2 | | | 3. Length of stay in ICU and hospital |
| Kawazoe et al. [37] (2017) | RCT | 101/ 100 | 69/68 | Yes | No | RASS<-2 | CAM-ICU | ①③ | 1. Mortality(28-day) |
| | | | | | | | | | 2. Ventilator-free days(28-day) |
| | | | | | | RASS-2 to 0 | | | 3. Sequential Organ Failure Assessment score (days 1, 2, 4, 6, 8) |
| | | | | | | | | | 4. Delirium and coma |
| | | | | | | | | | 5. Length of stay in ICU |
| | | | | | | | | | 6.Renal function, nutrition state, and inflammation |
| Kaplan et al. [38] (2018) | Before-after trial | 66/66 | 60/61 | Yes | Yes | RASS<-1 | CAM-ICU | ①②③ | 1. ventilator-free days(28-day) |
| | | | | | | | | | 2. Cumulative sedative requirements |
| | | | | | | RASS-1 to 1 | | | 3. level of sedation |
| | | | | | | | | | 4. Delirium |
| | | | | | | | | | 5. Length of stay in ICU and hospital |
| De Jonghe et al. [39] (2018) | RCT | 590/ 584 | 67/66 | Yes | No | RASS<-2* | CAM-ICU | ①②③ | 1. Mortality (28-day, 90-day, and 1-year) |
| | | | | | | RASS-2 to 0# | | | 2.Time to first spontaneous breathing trial |
| | | | | | | | | | 3.Time to successful extubation |

$N_D$: number in deeper sedation, $N_L$: number in lighter sedation, BDZ: benzodiazepine (Different BDZ means there was a significant difference in BDZ dosage between two groups), SI: Severity of Illness (Different SI means that the severity of illness was significantly different between two groups no matter what tools were used. MAAS: Motor Activity Assessment Scale, RASS: Richmond Agitation Sedation Scale, SAS: Sedation Agitation Scale. CAM-ICU: Confusion Assessment Method for the Intensive Care Unit, DSM-IV: Diagnostic and Statistical Manual of mental disorders-fourth edition.

* The median level of sedation at randomization (the usual care).

# Estimated with protocol of over-sedation prevention (the goal of intervention).

Endpoints: ①Delirium, ②Agitation-related adverse events, ③Mortality.

the results of 10 RCTs (2221 mechanically ventilated patients in total) still failed to demonstrate sedation depth as a risk factor for delirium in patients under MV (OR = 0.95, 95% CI: 0.79–1.13, $P$ = 0.548; Fig 4).

## Influence of confounding factors on the correlation between sedation depth and delirium occurrence

Thirteen of the 18 studies in this meta-analysis reported a significant difference in the dosage of benzodiazepine between the lighter and deeper sedation group ($P < 0.05$). In these 13 studies, the lighter sedation group had a lower dosage of benzodiazepines (S1 Table). Pooled forest plots revealed no significant difference in the delirium risk between different sedation depths among studies showing a significant difference in the dosage of benzodiazepines (OR = 0.77, 95% CI: 0.53–1.11, $P$ = 0.160; Fig 5). Three cohort studies [30,33,35] revealed a higher occurrence of delirium in the lighter sedation group with lower dosage of benzodiazepines than in

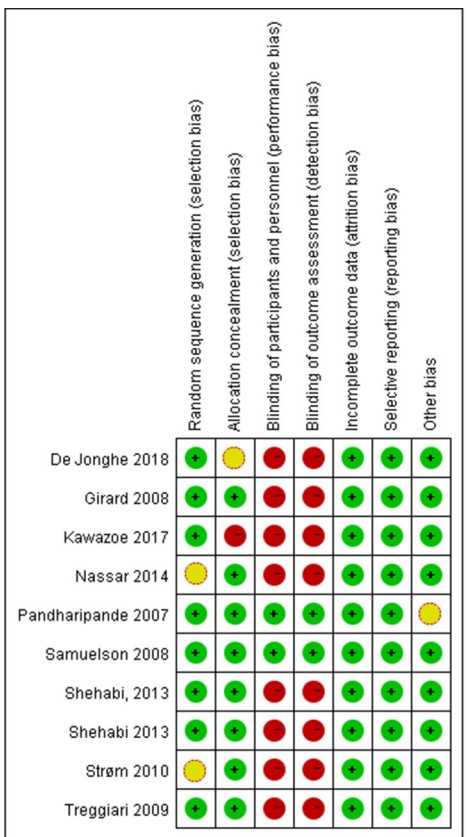

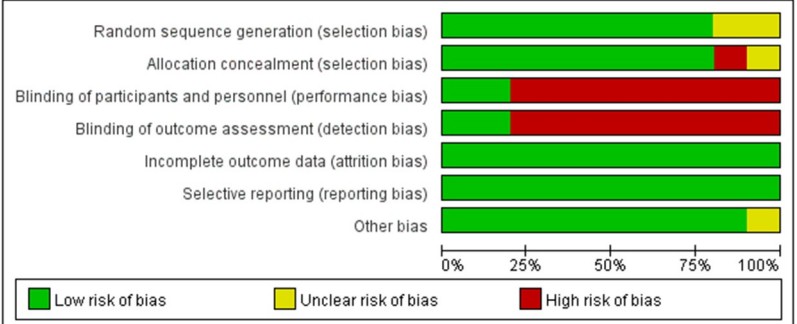

**Fig 2. Risk of bias graph.**

the deeper sedation group. These findings contrasted those of one study [34], reporting an increased occurrence of delirium in the deeper sedation group because of the increased dosage of benzodiazepines (Fig 5).

The pooled forest plots of the 13 studies, which reported no significant difference in disease severity between the two groups, interestingly revealed a significantly higher risk of delirium in the lighter sedation group (OR = 0.68, 95% CI: 0.49–0.94, $P$ = 0.021; Fig 6). Subgroup analysis was further conducted, based on whether a study had an RCT design. All three studies with a non-RCT design showed that light sedation increased the risk of delirium (OR = 0.40, 95% CI: 0.31–0.52, $P$ < 0.001; Fig 6). For remaining 10 RCTs, the meta-analysis failed to show that sedation depth significantly influenced the delirium occurrence in mechanically ventilated patients whoes disease severity did not differ (OR = 0.95, 95%CI: 0.79–1.13, $P$ = 0.548; Fig 6).

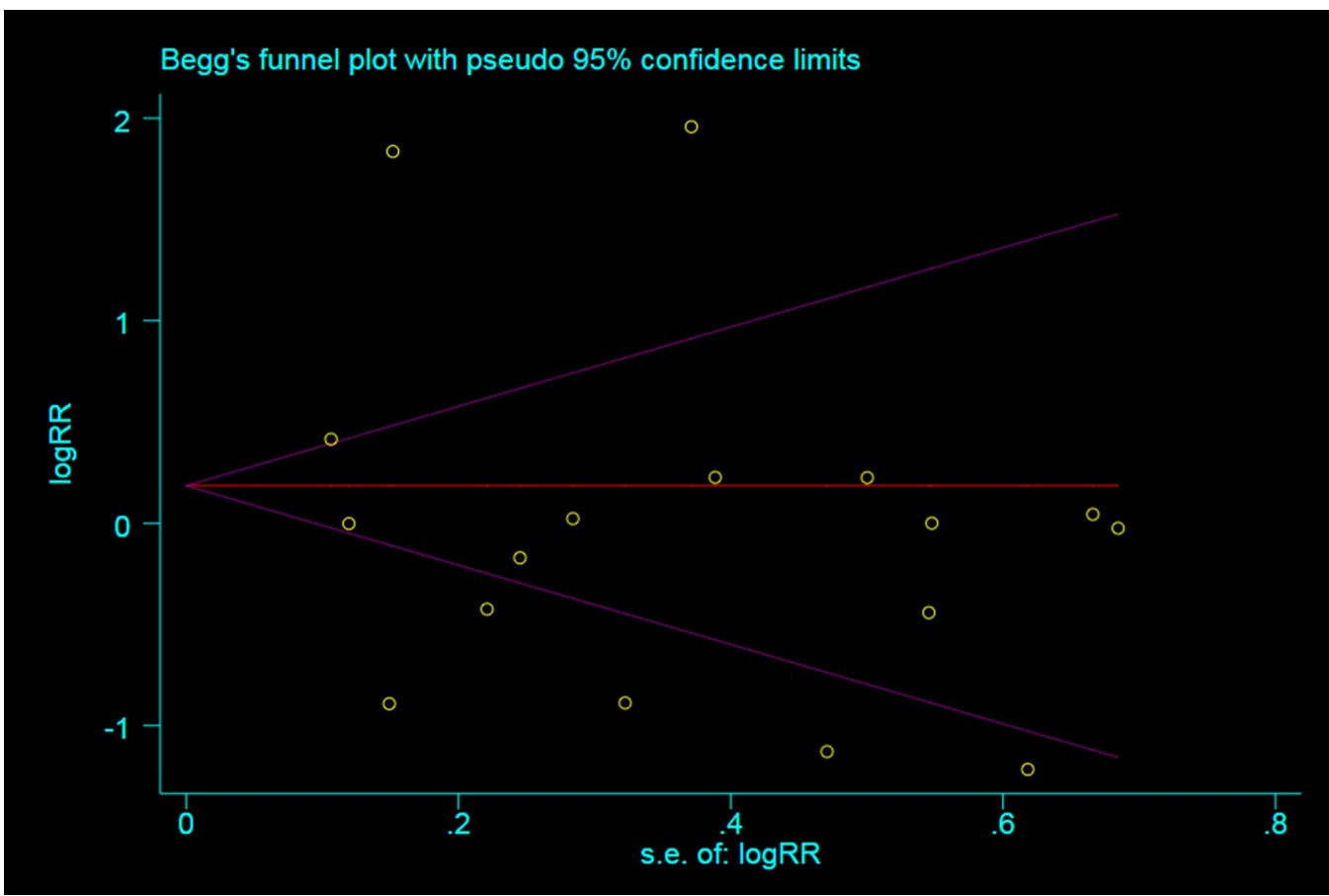

**Fig 3. Funnel plot of insertion success for publication bias detection.**

The risk of delirium significantly increased in parallel with higher dosages of benzodiazepines and greater disease severity in the deeper sedation group than in the lighter sedation group in Balzer's study (S1 Table and Fig 5) [34].

### Difference in sedation depth versus agitation-related adverse events and mortality

Nine of the 18 studies reported agitation-related adverse events, and included seven RCTs and two non-RCTs. Their findings (overall $I^2 = 0\%$) indicated a significantly higher risk of agitation-related adverse events in mechanically ventilated patients with lighter sedation (OR = 0.61, 95%CI: 0.45–0.84, $P = 0.002$; Fig 7).

Sixteen studies reported hospitalization mortality. Compared to the lighter sedation group, the deeper sedation group had a significantly higher risk of death (OR 1.82, 95% CI: 1.23–2.69, $P = 0.003$; Fig 8). In the subgroup analysis, nine RCTs and seven non-RCTs consistently indicated that deeper sedation significantly increased the risk of death (Fig 8).

### Results from Bayesian-based meta-analysis

Based on $I^2$ test analysis, heterogeneity was high (67.4% - 96.8%) among 18 studies that were included to confirm the effect of lighter sedation versus deeper sedation on delirium occurrence. The Bayesian method was consequently used for the meta-analysis. Bayesian random

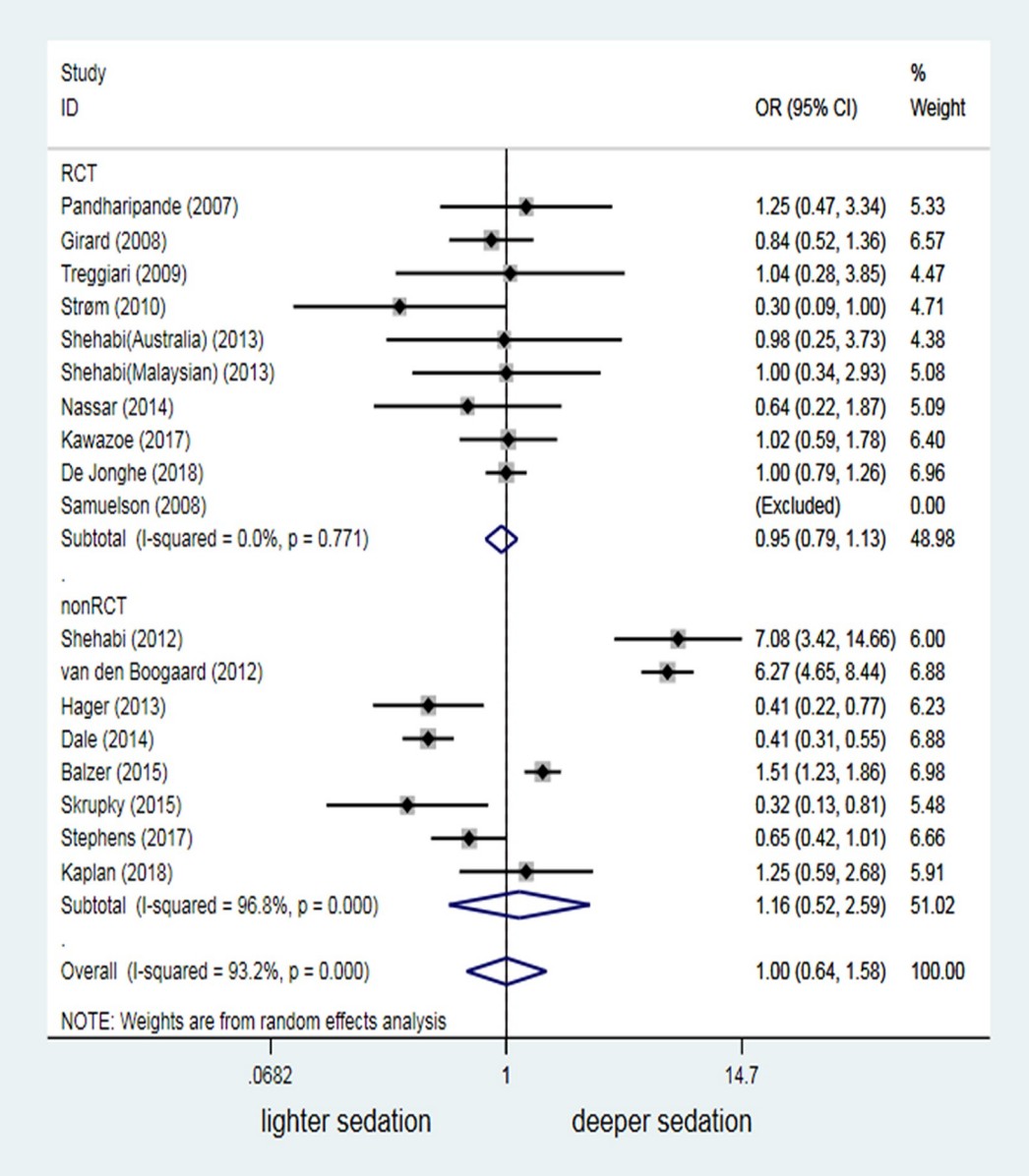

**Fig 4. Odds ratio of different sedation depth affecting occurrence of delirium.**

effect model showed an OR of 1.00 (95% CI: 0.61–1.62) for overall delirium occurrence, OR of 0.59 (95% CI: 0.35–0.95) for the incidence of agitation-related adverse events, and OR of 1.90 (95% CI: 1.27–2.85) for mortality comparisons, all of which showed the results consistent with those from traditional methods.

## Discussion

Unlike the findings of a previous meta-analysis [13], we enrolled more RCT studies that reported a relationship between sedation depth and delirium occurrence (2 studies with 97 mechanically ventilated patients in previous one vs. 10 studies with 2221 mechanically ventilated patients in our study). However, we could not conclude from our meta-analysis that

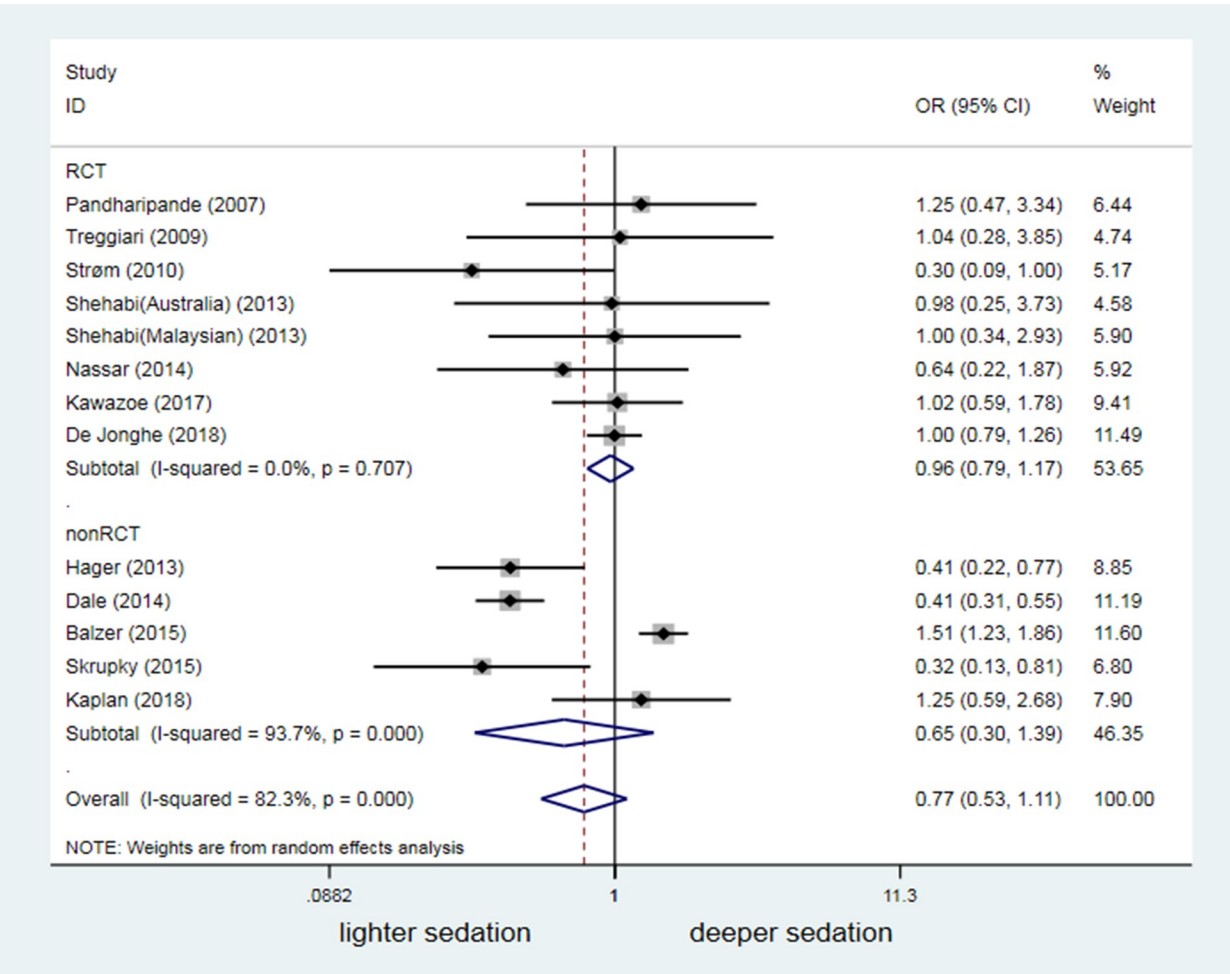

**Fig 5. Confounding effect of different BZD dosage on results of different sedation depth affecting occurrence of delirium.**
BZD = Benzodiazepines.

sedation depth altered the delirium risk in patients under MV (OR = 1.00, 95% CI: 0.64–1.58, *P* = 0.993; Fig 4). Similar to findings in previous studies [40,41], 16 merged analyses showed that deep sedation was a high risk factor for death in mechanically ventilated patients (OR = 1.82, 95% CI: 1.23–2.69, *P* = 0.003; Fig 8). However, our data suggested that lighter sedation significantly increased the potential risk of agitation-related adverse events in adult mechanically ventilated patients (OR = 0.61, 95%CI: 0.45–0.84, *P* = 0.002; Fig 7).

An increasing number of studies has examined the association of sedation depth with delirium occurrence in critically ill mechanically ventilated patients recently; however, few studies are RCTs. A systematic review of the literature indicated that light sedation (RASS ≥-2 or Ramsay 1 to 2) versus deep sedation (RASS≤-3 or Ramsay 3 to 4) was compared in only two RCTs that enrolled 37 and 60 mechanically ventilated patients respectively [19,31]. Most data comparing differences in sedation depth and delirium risk were obtained from non-RCT studies (e.g., prospective and retrospective cohort studies) [28,29,34,36]. Unfortunately, the studies' results were highly varied. We also found that three of eight cohort studies (3401 mechanically ventilated patients in total) showed a high risk of delirium with deeper sedation [28,29,34]. By contrast, another four cohort studies (2245 mechanically ventilated patients in total) suggested that lighter sedation increased the occurrence of delirium [30,33,35,36]. We noticed that a

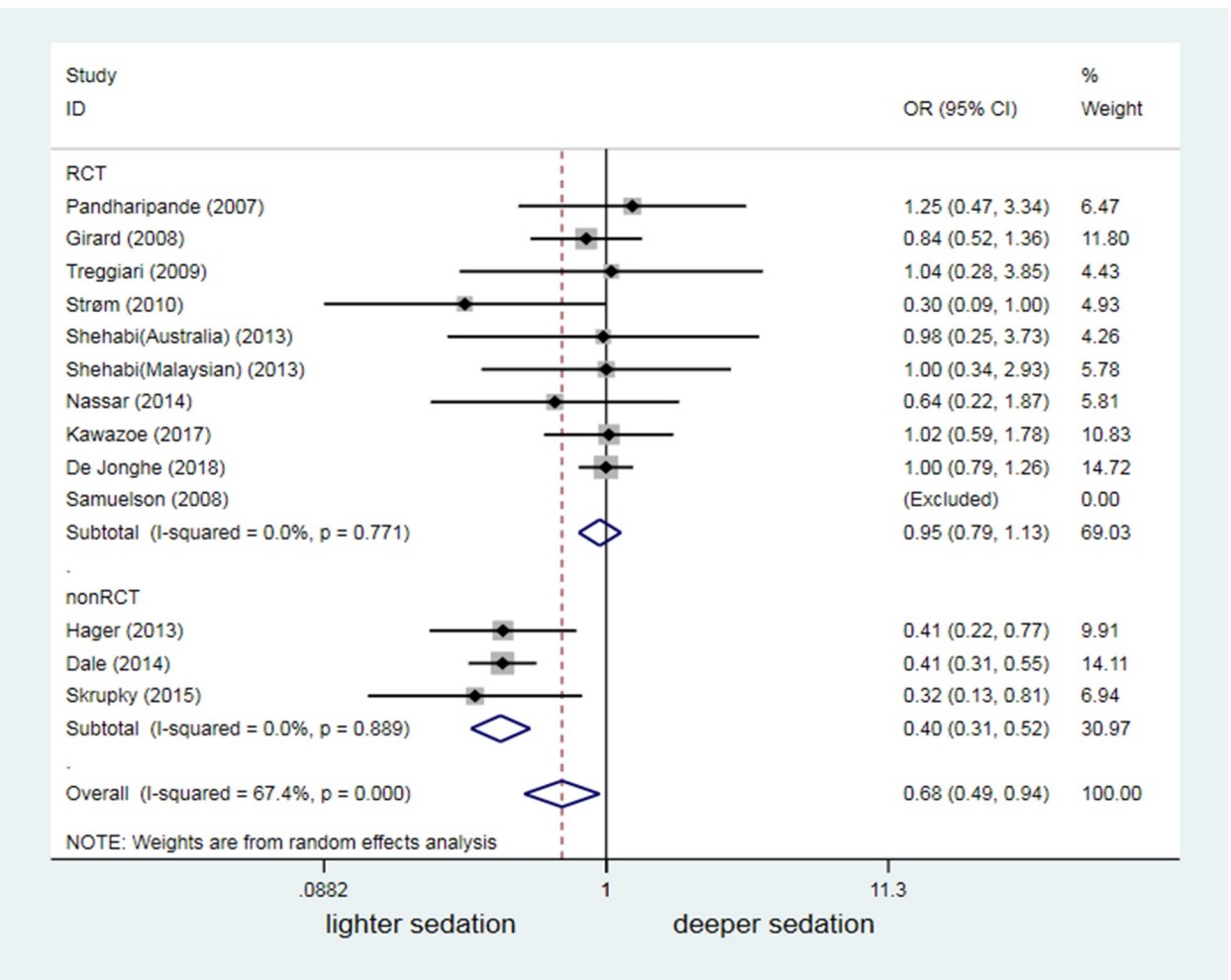

**Fig 6. Odds ratio of different sedation depth affecting occurrence of delirium in trials without difference in severity of diseases between the lighter and the deeper sedation groups.**

significantly different depth of sedation was documented between the intervention group and control group in some RCTs reporting delirium. However, these RCTs missed to enrolled into the previous meta-analysis because the intervention group versus the control group was not clearly categorized as light vs deep sedation. Therefore, we substituted the terms "lighter sedation" and "deeper sedation" for the terms "light sedation" and "deep sedation" in the literature search strategy. Importantly, it was aimed to emphasize the effect of differences in sedation depth on occurrence of delirium and to avoid the confounding effect of different definitions of light and deep sedation between studies. Successfully, 10 RCTs were enrolled in this meta-analysis. However, RCTs reporting the occurrence of delirium as the primary or secondary outcome had a lower heterogeneity and did not provide conclusive information that sedation depth might alter the delirium risk in mechanically ventilated patients. Therefore, any unilateral emphasis on the results from low-quality studies (e.g., deep or light sedation may increase delirium risk) should be avoided.

Use of benzodiazepines was highly concerned as a risk factor for delirium in mechanically ventilated patients [42–44], which might confound the association between sedation depth and risk of delirium. But, controversy remained owing to highly heterogeneous quality of those studies [45]. Our subgroup analysis similarly suggested contrasting findings among non-

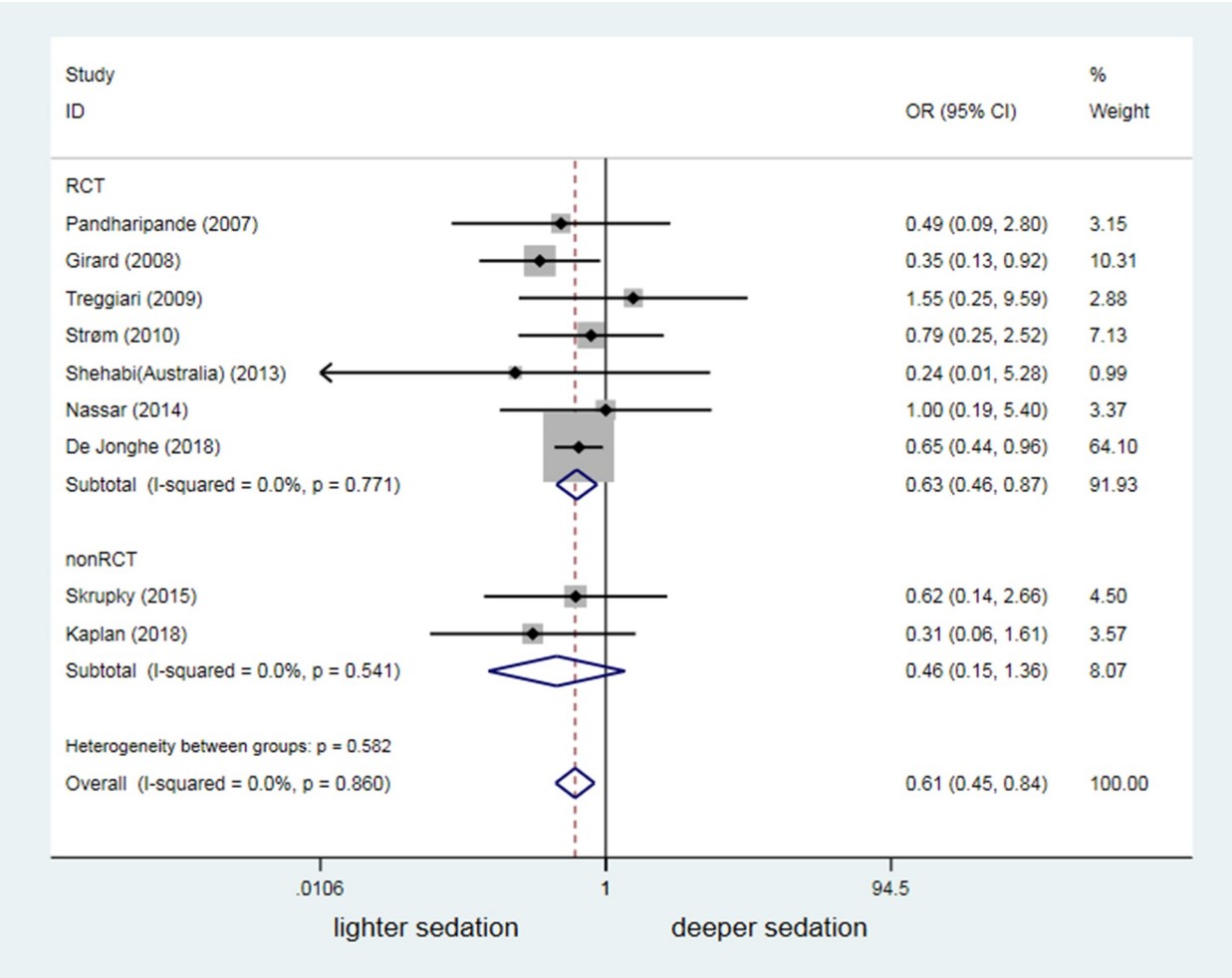

**Fig 7. The impact of discrepancy in depth of sedation on frequency of agitation-related adverse events.**

RCTs (Fig 5). However, the pooled forest plots of eight RCTs demonstrated that differences in the benzodiazepine dosage did not alter the conclusion that sedation depth does not influence the risk of delirium (OR = 0.77, 95% CI: 0.53–1.11, $P = 0.160$; Fig 5). Another concern regarding a confounding factor was the disease severity. Pandharipande et al. [42] reported that mechanically ventilated patients with a high score on the Acute Physiology and Chronic Health Evaluation II (APACHEII) scale or Sequential Organ Failure Assessment (SOFA) scale had a significantly increased risk of delirium. Thirteen of the 18 studies included in this meta-analysis interestingly reported no significant difference in disease severity between the two groups (Additional S1 File). Results from 10 RCTs (with higher quality) of these 13 studies suggested that the sedation depth did not change the risk of delirium in mechanically ventilated patients. By contrast, pooled forest plots of subgroup analysis from the other three non-RCTs studies showed that light sedation augmented the delirium risk (OR = 0.40, 95% CI: 0.31–0.52, $P<0.001$). After adjusting for the potential confounders benzodiazepine use and disease severity, our data remained inconclusive regarding for the association between sedation depth and risk of delirium in mechanically ventilated patients. Therefore, further studies are needed.

This meta-analysis also focused on sedation depth in relation to the risk of agitation-related adverse events and mortality in the enrolled studies. Consistent with the meta-analysis by

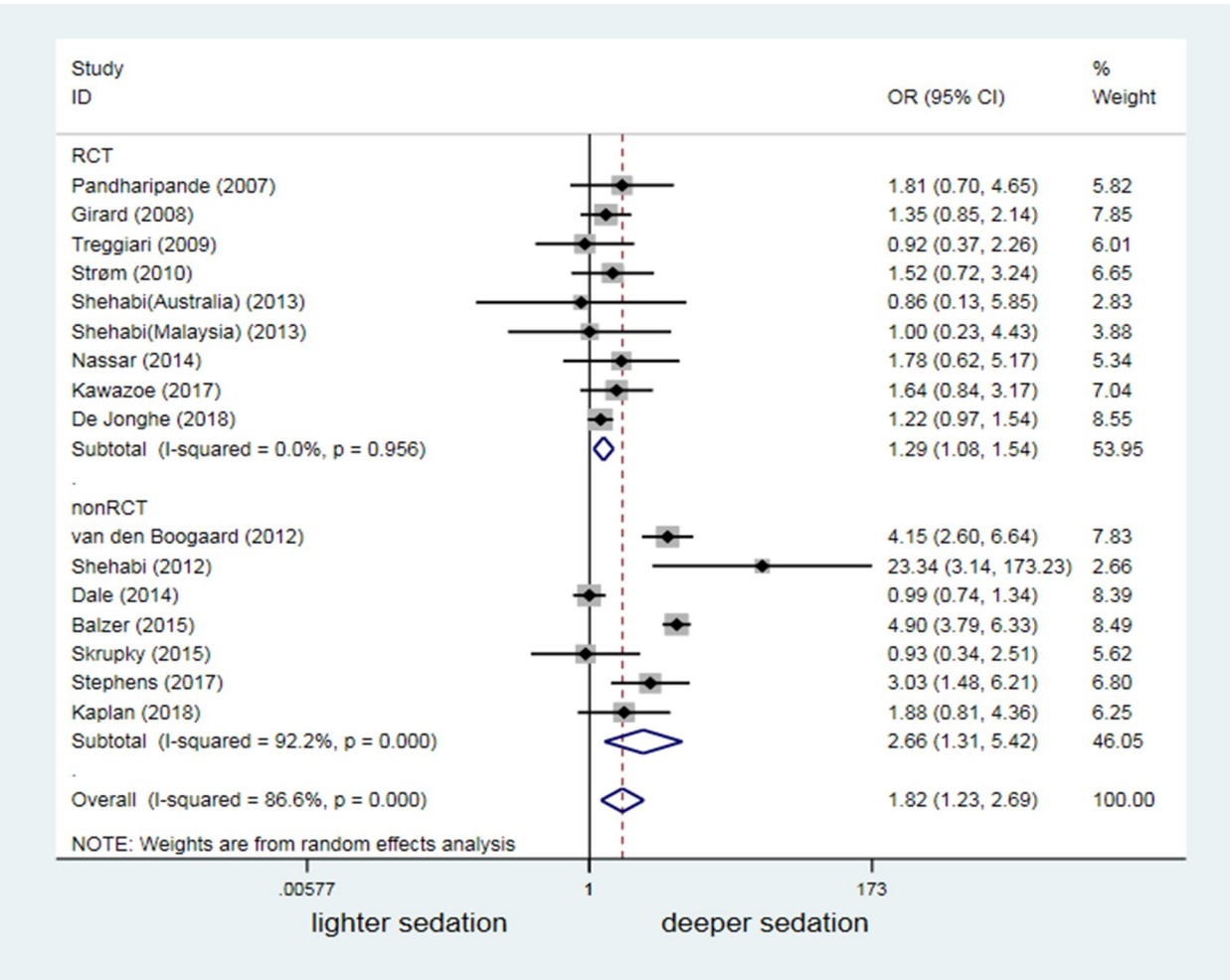

**Fig 8. The impact of discrepancy in depth of sedation on mortality.**

Stephens et al. [13], which was that it focused on the first 48 hours of mechanical ventilation whereas our study includes studies across the course of critical illness. Our findings further convinced that deep sedation was a high risk factor for poor prognosis, based on a bigger sample size from studies with high quality. These results will inevitably strengthen the importance of maintaining light levels of sedation for most of adult ICU patients under MV, as recommended by the updated guidelines [46–48]. However, we also found that lightly sedated mechanically ventilated patients might face a significantly high risk of agitation-related adverse events (OR = 0.61, 95% CI: 0.45–0.84, P = 0.002; Fig 7). Therefore, the challenge remains to our sedation practices either avoiding unjustified deep sedation or preventing MV patients from agitation and agitation-related adverse events [39,49,50].

This systematic review had several limitations. First, we enrolled RCTs and non-RCTs. The association between sedation depth and risk of delirium was significant but in opposite directions based on two parts of these non-RCTs. Although a high bias risk was not identified by sensitivity analysis, the quality of these studies might contribute to the controversy results largely. Second, the enrolled studies were inconsistent in assessing sedation depth (e.g., MAAS, RASS, SAS and Ramsay), evaluating severity of disease (e.g., APACHE II, Simplified Acute Physiology Score II, and SOFA), and diagnosing delirium (e.g., CAM-ICU and

DSM-IV). However, the tools used the same in the two groups in each study. Notably, we focused on the effect of different sedation depths (i.e. "lighter sedation" vs "deeper sedation" rather than "light sedation" vs "deep sedation") on the risk of delirium (both assessments were accepted widely) in mechanically ventilated patients. Therefore, the results are reliable. Third, statistical analyses revealed high heterogeneity, which might be related to the two limitations mentioned above. We managed to reduce this bias through subgroup analysis and a Bayesian method. We found that the results of most subgroup analyses seemed similar to those of the general meta-analysis.

## Conclusions

In this systematic review of available literature and a meta-analysis, findings were inconclusive that different sedation depth would change the risk of delirium in critically ill patients with MV. An increased risk of death was associated with deeper sedation, whereas lighter sedation might have a risk potential for agitation-related adverse events. Therefore, further researches are needed to optimize sedation strategy for critically ill adult patients receiving MV.

## Supporting information

**S1 Checklist. PRISMA checklist.**
(DOC)

**S1 Fig. Newcastle Ottawa scale quality assessment.** ICU: Intensive Care Unit, APACHE II: Acute Physiology and Chronic Health Evaluation II, SAPS II: Simplified Acute Physiology Score II, BMI: Body Mass Index, ED: emergency department.
(TIF)

**S2 Fig. Diagram of insertion success for sensitivity analysis.**
(TIF)

**S1 Table. Benzodiazepine used and patients' severity of illness in the included studies.** D/L: Deeper/Lighter gourp, BDZ: benzodiazepine (Different BDZ means there was a significant difference in BDZ dosage between two groups), 1: mg/h, 2: mg/d, 3: mg(cumulative dose) 4: mg/kg/h, 5: mg/kg(cumulative dose), 6: %(% of patient midazolam was given), -: means without BZD or data unavailable. a: APACHEⅡ(Acute Physiology and Chronic Health EvaluationⅡ), b: SAPSⅡ (Simplified Acute Physiology ScoreⅡ), C: SOFA (Sequential Organ Failure Assessment)
(DOC)

**S2 Table. Cochrane bias assessment of randomized studies.**
(DOC)

**S1 File. Search strategy.**
(DOC)

**S2 File. The table of GRADE.**
(DOCX)

**S3 File. The list of excluded studies.**
(DOCX)

## Acknowledgments

All authors contributed to the critical review and revision of the manuscript. All authors have seen and approved the final version of the manuscript.

## Author Contributions

**Conceptualization:** Heling Zhao, Penglin Ma.

**Data curation:** Ling Long, Shan Ren, Yichun Gong, Haotian Zhao.

**Formal analysis:** Ling Long, Haotian Zhao, Penglin Ma.

**Methodology:** Limin Shen.

**Supervision:** Heling Zhao, Penglin Ma.

**Writing – review & editing:** Ling Long, Cong He.

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
