## [Decision Letter · Decision Letter 0]

23 Jan 2020

PONE-D-19-33669

Different Depths of Sedation Versus Risk of Delirium in Adult Mechanically Ventilated Patients: A Systematic Review and Meta-Analysis

PLOS ONE

Dear Dr Ma,

Thank you for submitting your manuscript to PLOS ONE. After careful consideration, we feel that it has merit but does not fully meet PLOS ONE’s publication criteria as it currently stands. Therefore, we invite you to submit a revised version of the manuscript that addresses the points raised during the review process.

We would appreciate receiving your revised manuscript by Mar 08 2020 11:59PM. To enhance the reproducibility of your results, we recommend that if applicable you deposit your laboratory protocols in protocols.io, where a protocol can be assigned its own identifier (DOI) such that it can be cited independently in the future. For instructions see: http://journals.plos.org/plosone/s/submission-guidelines#loc-laboratory-protocols

We look forward to receiving your revised manuscript.

Kind regards,

Luciane Cruz Lopes, PhD

Academic Editor

PLOS ONE

Journal Requirements:

2. We note that the original search was performed in December 2019. To ensure that your systematic review is up-to-date and comprehensive, we ask that you please update the search period to ensure that the literature search includes studies that have been published within the last 12 months.

3. At this time, we ask that you please move the Newastle-Ottawa scale quality assessment, risk of bias graph and funnel plot provided as additional files in your submission to the manuscript as Figure files.

5. Please include your tables as part of your main manuscript and remove the individual files. Please note that supplementary tables (should remain/ be uploaded) as separate "supporting information" files.

6. Thank you for stating the following financial disclosure:

a) Please provide an amended Funding Statement that declares *all* the funding or sources of support received during this specific study (whether external or internal to your organization) as detailed online in our guide for authors at http://journals.plos.org/plosone/s/submit-now.  

b) Please state what role the funders took in the study.  If any authors received a salary from any of your funders, please state which authors and which funder. If the funders had no role, please state: "The funders had no role in study design, data collection and analysis, decision to publish, or preparation of the manuscript."

7. Thank you for stating the following in your Competing Interests section:  

Additional Editor Comments:

Dear Authors,

        My assessment is very similar to the two reviewers. I think that paper is described but it needs to add some important analyzes, for example the quality analysis of the body of evidence separately for each outcome (GRADE) as mentioned by one of the reviewers.

   I also didn't see the discussion about sensitivity and heterogeneity analysis.

I also agree with the lack of clarity about the exposure variable definition of depth of sedation with different cut-offs for sedation depths among the included studies.

It would be great if authros could answer point by point suggested by reviewers

Reviewers' comments:

Reviewer's Responses to Questions

**Comments to the Author**

1. Is the manuscript technically sound, and do the data support the conclusions?

Reviewer #1: Yes

Reviewer #2: Yes

2. Has the statistical analysis been performed appropriately and rigorously? 

Reviewer #1: Yes

Reviewer #2: Yes

3. Have the authors made all data underlying the findings in their manuscript fully available?

Reviewer #1: Yes

Reviewer #2: Yes

4. Is the manuscript presented in an intelligible fashion and written in standard English?

Reviewer #1: Yes

Reviewer #2: Yes

5. Review Comments to the Author

Reviewer #1: The manuscript “Different Depths of Sedation Versus Risk of Delirium in Adult Mechanically Ventilated Patients: A Systematic Review and Meta-Analysis” by Ma and colleagues sought to answer the question whether different sedation depths are associated with higher delirium risk. Based on their findings, they concluded that at present, there is inconclusive evidence whether different sedation depths would change the risk of delirium in adult mechanically ventilated patients. Overall, the manuscript is drafted appropriately, and the authors have looked through the confounding factors that may affect the outcome. I have some comments which are outlined below.

1. I am a little concerned about the exposure variable definition of depth of sedation. As different studies used different cut-offs for sedation depths, sweeping statement based on combining these different exposures may lead us to inconclusive findings as seen in the review. That will be my major concern while trying to conclude depths of sedation as a risk factor for delirium when the exposure depths are not same in the studies. While most studies seem to be using the cut-off of RASS greater than or equal to -3, it is not the same with all the studies. It may be worthwhile to pool studies which are consistent and see if the findings hold.

2. The other major concern I have is regarding the outcome variable which is delirium incidence. How is this defined? In the literature, it has been defined multiple ways and I am not sure how the authors have construed it for their review. This again will result in discrepancy among studies. As there has been debate about how to define incidence versus prevalence especially in ICU settings where patients may have been admitted with delirium, the authors may want to think this over. One suggestion is to keep it general with a term such as delirium occurrence.

3. An interesting finding to note is the absence of a confounding effect of BZD on delirium. This really needs to be highlighted in the discussion. BZD have been thought of as risk factors for delirium, but the authors findings point towards a much-nuanced issue. Are these effects independent of sedation depth or is it a direct effect? The authors need to comment further on this in the discussion.

4. The findings of light sedation increasing delirium risk need to be further elaborated as this seems counter intuitive to current approaches.

5. Based on the mortality findings, in the absence of increasing delirium risk, depth of sedation is independently associated with mortality. Is this cause or could this be due to sicker patients kept deeply sedated. I would suspect that as this finding holds in RCT with randomized approach, the latter is not the case. This will need further discussion in the discussion section. May be an analysis with stratification based on illness severity may help clarify this further.

Reviewer #2: Although the authors did not assess the quality of the evidence set and the strength of the recommendation (GRADE), for each of the results analyzed, I think that the review was very well conducted, that the problem of delusion is the order of the day in intensive care units and, why, the results of the studies may help a lot in making decisions about sedation and analgesia of patients in these units.

6. PLOS authors have the option to publish the peer review history of their article (what does this mean?). If published, this will include your full peer review and any attached files.

Reviewer #1: No

Reviewer #2: No

---

## [Author Response · Author response to Decision Letter 0]

20 Feb 2020

Thank you for letter and for the reviewer’ comments concerning our manuscript. We have responded all the comments in the file that named by Response to Reviewers.

---

## [Decision Letter · Decision Letter 1]

18 May 2020

PONE-D-19-33669R1

Different Depths of Sedation Versus Risk of Delirium in Adult Mechanically Ventilated Patients: A Systematic Review and Meta-Analysis

PLOS ONE

Dear Dr Ma,

Thank you for submitting your manuscript to PLOS ONE. After careful consideration, we feel that it has merit but does not fully meet PLOS ONE’s publication criteria as it currently stands. Therefore, we invite you to submit a revised version of the manuscript that addresses the points raised during the review process.

We would appreciate receiving your revised manuscript by Jul 02 2020 11:59PM. To enhance the reproducibility of your results, we recommend that if applicable you deposit your laboratory protocols in protocols.io, where a protocol can be assigned its own identifier (DOI) such that it can be cited independently in the future. For instructions see: http://journals.plos.org/plosone/s/submission-guidelines#loc-laboratory-protocols

We look forward to receiving your revised manuscript.

Kind regards,

Luciane Cruz Lopes, PhD

Academic Editor

PLOS ONE

Reviewers' comments:

Reviewer's Responses to Questions

**Comments to the Author**

1. If the authors have adequately addressed your comments raised in a previous round of review and you feel that this manuscript is now acceptable for publication, you may indicate that here to bypass the “Comments to the Author” section, enter your conflict of interest statement in the “Confidential to Editor” section, and submit your "Accept" recommendation.

Reviewer #2: All comments have been addressed

Reviewer #3: (No Response)

2. Is the manuscript technically sound, and do the data support the conclusions?

Reviewer #2: Yes

Reviewer #3: Yes

3. Has the statistical analysis been performed appropriately and rigorously? 

Reviewer #2: Yes

Reviewer #3: Yes

4. Have the authors made all data underlying the findings in their manuscript fully available?

Reviewer #2: Yes

Reviewer #3: Yes

5. Is the manuscript presented in an intelligible fashion and written in standard English?

Reviewer #2: Yes

Reviewer #3: Yes

6. Review Comments to the Author

Reviewer #2: I did not notice any problems in the article, therefore, I recommend that it be published as soon as possible, in view of its timeliness and relevance to the scientific community, especially in the field of medicine.

Reviewer #3: With great interest I read the adapted version of the manuscript. The authors addressed the comments of the previous reviewers adequately which led to significant improvement of the research and the manuscript. The authors did a rigorous meta-analysis and show great adherence to existing standards and protocols in place for such research. Doing so, the research is up to current standards and very suitable for publication in Plos One. And although the manuscript is almost in its final version I do have some concerns reading the research for the first time.

My main concern is the fact that the relationship between sedation (depth) and delirium is very complex and influenced by many factors. The effort of the authors to try to include observational studies in their meta analysis is a great effort but introduces the bias of observational research in their meta analysis. Observational research in ICU research is prone to bias and relies on advanced statistical methods including time-varying variables, competing events, confounding by indication, and addressing immortal time bias. T

For example, there is the challenge that light sedation levels interfere with delirium diagnosis and comparing them with deep sedation give therfore regression to the mean (for example patient scoring RASS-2 is CAM-ICU positive (item 1, 2 and 4 present)).

Or the problem that disease severity and dosage of sedatives are not pure confounding factors as both cause deep sedation (placing them in causal chain of high severity of illnessdeep sedationdelirium). This causal chain is also seen in the risk of deep sedation and risk of mortaltiy; deep sedation being a proxy for severity of illness and thus not surprisingly being associated with higher mortality.

Or the challenge that only patients in light sedation have the oppurtunity to develop adverse events (immortal time biase..you have to be awake enough to remove the ET).

I feel that the authors do address a lot or these issues by reporting the results separetly for RCTs and observational research but I would appreciate it if the authors emphasize the risk of bias of observational research and thus of the reported overall ORs. For example, the overall ORs on the relationship of mortaltiy and adverse events seem to be driven by the observational research.

7. PLOS authors have the option to publish the peer review history of their article (what does this mean?). If published, this will include your full peer review and any attached files.

Reviewer #2: No

Reviewer #3: No

---

## [Author Response · Author response to Decision Letter 1]

27 May 2020

Thanks again for your comments. Indeed, the relationship between the depth of sedation and delirium is complex and influenced by many factors. The severity of the disease and the dosage of benzodiazepines are considered to be important factors. Therefore, we conducted a subgroup analysis of these factors and found that differences in the benzodiazepine dosage did not alter the conclusion that sedation depth does not influence the risk of delirium. But the patients with the same severity of the disease have the higher risk of delirium with the lighter sedation(OR = 0.68, 95% CI: 0.49-0.94, P = 0.021) and higher risk of mortality with the deeper sedation(OR = 1.20, 95% CI: 1.03-1.39, P = 0.020). So, the severity of illness seemed to confound this conclusion less likely. 

Our systematic review included some observational research, which generated high heterogeneity. In order to reduce the weight of observational research in the overall results, we used a Bayesian method. It is found that the results of most subgroup analyses seemed similar to those of the general meta-analysis. In terms of the risk of mortality and incidence of adverse events, the ORs of the RCTs, observational research and overall research were all on the same side of the forest plots. Nevertheless, we agree with and appreciate the comments of reviewer #3. We have stressed these views in the discussion of the revised manuscript. "Due to the methodological limitations, we should carefully interpret the meta results of observational research. It is evident that the likelihood of diagnosing delirium and developing adverse events is higher in patients sedated with the lighter depth, while the effect of deep sedation on mortality needs to be adjusted with factors such as disease severity. " (shown on page 13, line 330 to 336, marked in red.)

---

## [Editor Report · Decision Letter 2]

12 Jun 2020

PONE-D-19-33669R2

Different Depths of Sedation Versus Risk of Delirium in Adult Mechanically Ventilated Patients: A Systematic Review and Meta-Analysis

PLOS ONE

Dear Dr. Ma,

Thank you for submitting your manuscript to PLOS ONE. After careful consideration, we feel that it has merit but does not fully meet PLOS ONE’s publication criteria as it currently stands. Therefore, we invite you to submit a revised version of the manuscript that addresses the points raised during the review process.

We look forward to receiving your revised manuscript.

Kind regards,

Luciane Cruz Lopes, PhD

Academic Editor

PLOS ONE

Additional Editor Comments (if provided):

Dear Authors,

Please, could you address the follow topic below?

1) To explain their selection of the study designs for inclusion in the review in method section - Explanation for including both RCTs and NRSI

2) To provide a list of excluded studies and justify the exclusions

3) To report on the sources of funding for the studies included in the review

---

## [Author Response · Author response to Decision Letter 2]

23 Jun 2020

A1: There is a lack of research published RCTs primarily regarding the effect of sedation depth on delirium occurrence in critically ill patients with MV and more studies on the outcome of delirium are cohort study. Therefore, it is suggested that the inclusion of the cohort study may increase the information of systematic review on the exploration of delirium risk factors, but the lower quality of data is a fact. The RCTs and NRSI must meet the PICO principle. The exclusion criteria have been clarified in the methods section in this paper(marked in blue).

A2 and A3: Thank you for your comments. The list of excluded studies and the sources of funding for the studies included in the review has been uploaded as S3 File and S4 File respectively.

---

## [Editor Report · Decision Letter 3]

29 Jun 2020

Different Depths of Sedation Versus Risk of Delirium in Adult Mechanically Ventilated Patients: A Systematic Review and Meta-Analysis

PONE-D-19-33669R3

Dear Dr. Ma,

We’re pleased to inform you that your manuscript has been judged scientifically suitable for publication and will be formally accepted for publication once it meets all outstanding technical requirements.

Kind regards,

Luciane Cruz Lopes, PhD

Academic Editor

PLOS ONE

Additional Editor Comments (optional):

Dear author,

Could you please, check the quality of the figures and review all tables? It is necessary to improve the quality of the presentation and format. Please, review all references checking if the citation is missing. Thank you.
---

## [Editor Report · Acceptance letter]

1 Jul 2020

PONE-D-19-33669R3 

Different Depths of Sedation Versus Risk of Delirium in Adult Mechanically Ventilated Patients: A Systematic Review and Meta-Analysis 

Dear Dr. Ma:

I'm pleased to inform you that your manuscript has been deemed suitable for publication in PLOS ONE. Congratulations! Your manuscript is now with our production department. 

Kind regards, 

on behalf of

Dr. Luciane Cruz Lopes 

Academic Editor

PLOS ONE